# Contextual Constraints: Dynamic Evolution of Snake Venom Phospholipase A_2_

**DOI:** 10.3390/toxins14060420

**Published:** 2022-06-20

**Authors:** Vivek Suranse, Timothy N. W. Jackson, Kartik Sunagar

**Affiliations:** 1Evolutionary Venomics Laboratory, Centre for Ecological Sciences, Indian Institute of Science, Bangalore 560012, India; viveksuranse@iisc.ac.in; 2Australian Venom Research Unit, Department of Pharmacology and Therapeutics, The University of Melbourne, Parkville, VIC 3010, Australia; timothy.jackson@unimelb.edu.au

**Keywords:** phospholipase A_2_, snake venom, venom evolution, venom ecology, Elapidae, Viperidae

## Abstract

Venom is a dynamic trait that has contributed to the success of numerous organismal lineages. Predominantly composed of proteins, these complex cocktails are deployed for predation and/or self-defence. Many non-toxic physiological proteins have been convergently and recurrently recruited by venomous animals into their toxin arsenal. Phospholipase A_2_ (PLA_2_) is one such protein and features in the venoms of many organisms across the animal kingdom, including snakes of the families Elapidae and Viperidae. Understanding the evolutionary history of this superfamily would therefore provide insight into the origin and diversification of venom toxins and the evolution of novelty more broadly. The literature is replete with studies that have identified diversifying selection as the sole influence on PLA_2_ evolution. However, these studies have largely neglected the structural/functional constraints on PLA_2_s, and the ecology and evolutionary histories of the diverse snake lineages that produce them. By considering these crucial factors and employing evolutionary analyses integrated with a schema for the classification of PLA_2_s, we uncovered lineage-specific differences in selection regimes. Thus, our work provides novel insights into the evolution of this major snake venom toxin superfamily and underscores the importance of considering the influence of evolutionary and ecological contexts on molecular evolution.

## 1. Introduction

Venoms are functional traits used by one organism to subdue or deter another—their function is thus intrinsically ecological, as it mediates interactions between two organisms [1]. Venoms are secretory cocktails composed of a variety of ‘exophysiological’ proteins and peptides (collectively ‘toxins’) that interfere with the functioning of target molecules in key regulatory pathways [1,2,3]. Snake venoms, for example, are constituted by several toxin superfamilies, including phospholipase A_2_ (PLA_2_), three-finger toxins (3FTx), snake venom metalloproteinases (SVMP), snake venom serine proteases (SVSP), lectins, L-amino acid oxidase (LAAO), hyaluronidase, and Kunitz-type serine protease inhibitors [4]. While some of the major toxin superfamilies are especially exploited by specific lineages of snakes (e.g., 3FTx by elapids and SVMPs by viperids), others, such as PLA_2_s, are ubiquitously found in advanced snakes. Understanding the evolutionary origin and diversification of toxin superfamilies and their role in underpinning the evolutionary success of snakes is therefore intriguing.

PLA_2_s belong to a family of proteins that catalyse the hydrolysis of phospholipids by cleaving the 2-acyl ester linkage of 3-sn-phospholipids in a calcium-dependent manner [5,6]. In mammals, PLA_2_s play a pivotal role in crucial physiological processes, including fertilisation [7], cell proliferation [8], lipid metabolism [9] and signal transduction [10]. PLA_2_s are also amongst the major components of snake venoms, where they exhibit diverse biochemical activities including cytotoxicity, myotoxicity, neurotoxicity, inflammation and anticoagulatory effects [11].

Evidence has generally been consistent with the hypothesis that genes encoding snake venom toxins evolve under the influence of positive selection [12,13,14,15,16]. Venom PLA_2_s are no exception [15,17,18,19]. Unfortunately, however, studies investigating the nature of the selection pressures sculpting the evolution of PLA_2_s have failed to incorporate the unique evolutionary histories of snakes, including the ecological constraints experienced by them, as well as the structural and functional constraints on PLA_2_ toxins. These influences have been previously shown to constrain the evolutionary rates of other prominent snake venom toxin superfamilies, such as three-finger toxins (3FTx) of Elapidae snakes [14].

Snake venom PLA_2_s are classified into group I (Elapidae snakes) and group II (Viperidae snakes) based on their structural similarities with mammalian pancreatic and synovial PLA_2_s, respectively [20,21,22]. Group I PLA_2_s are further subclassified into pancreatic and non-pancreatic sequences based on the presence or absence of the pancreatic loop, a five amino acid motif. Given their medical relevance, snake venom PLA_2_s have been extensively studied over the past few decades. The two groups of PLA_2_ were independently ‘recruited’ as snake venom toxins. Recruitment of group II as a toxin appears to have occurred in the viperid snake lineage as a result of the co-option of a gene copy that was previously constitutively expressed in the venom gland [23]. Subsequent to recruitment, the family has expanded and contracted in various viperid lineages [24]. The evolutionary history of group I has not been reconstructed with the same level of detail; however, current literature is in broad agreement that snake venom PLA_2_s in general have diversified via a ‘birth-and-death’ model of evolution under the influence of positive selection [15,17,18,19,25,26]. However, both structural–functional constraints on the molecular evolution of PLA_2_s and the ecology and evolutionary history of snakes that produce them likely contribute to shaping the evolution of this toxin superfamily. Consideration of these factors has been largely neglected. Failure to incorporate these considerations hampers our understanding of the evolution of snake venom and could potentially lead to erroneous conclusions.

To address these shortcomings, we assembled a dataset of 912 sequences and classified them into various structurally, ecologically, and evolutionarily appropriate subgroups (Table 1; Appendix A). We then quantified the influence of selection on the evolution of this snake venom superfamily, utilising a range of bioinformatic analyses. Our findings highlight the dynamic evolution of snake venom PLA_2_s and clearly demonstrate the need to investigate the evolutionary histories of venom proteins in the light of their structural and functional constraints and the ecological and evolutionary contexts of the organisms that produce them.

## 2. Results and Discussion

### 2.1. Structural and Functional Underpinnings of Snake Venom PLA_2_s

Partitioning of datasets in this study is based on a variety of considerations. For example, β-bungarotoxins, which are found in the venoms of kraits (Elapidae: *Bungarus*), provide a vivid example of the way in which the structure–function relationships of molecules can constrain their evolution. β-bungarotoxins are chimeric toxins formed by the dimerisation of phospholipase A_2_ and Kunitz peptide subunits [27,28]. For the chains to dimerise and produce an active toxin, it is imperative to conserve contact residues that are involved in dimerisation. Hence, due to this unique constraint, these toxins were considered a distinct dataset. On the other hand, certain group I PLA_2_s are known to possess a characteristic five-amino-acid-long pancreatic loop [29]. Based on the presence or absence of this structural feature, we divided these PLA_2_s into pancreatic and non-pancreatic PLA_2_s, respectively ([20]; Appendix A).

In group II PLA_2_s of Viperidae snakes, the 49th amino acid residue of the mature toxin chain is responsible for the catalytic activity [30,31]. The presence of aspartic acid (D) at this position is known to impart catalytic activity, and the substitution of this amino acid for lysine (K) results in a complete loss of phospholipase activity. Several other substitutions have also been reported at this position [32,33,34]. Based on the amino acid substitution observed at this position, viperid PLA_2_s (a total of 494 sequences) were classified as D-, K-, N- or S-type datasets (Appendix A). We were also able to retrieve H-, R- and T-type sequences. However, since these sequences were in limited numbers, these datasets were excluded from analyses.

### 2.2. The Influence of Diverse Ecologies on the Evolution of Snake Venom PLA_2_s

We considered the diverse ecological contexts in which snakes are embedded while delineating PLA_2_ datasets. For example, the venom system of the marbled sea snake (*Aipysurus eydouxii*)—a lineage that has experienced a significant shift in feeding ecology from actively hunting fish to feeding on their eggs (oophagy)—has undergone considerable degeneration [35]. When toxin encoding genes were investigated in this species, it was found that the only 3FTx genes still expressed had undergone a frameshift mutation, rendering them dysfunctional [36]. A previous study, when investigating the evolution of the PLA_2_ genes in this species, reported a decelerated rate of evolution [37]. However, the methodology that was implemented to assess the influence of selection may have led to misleading results. Hence, we partitioned *A. eydouxii* PLA_2_s as a distinct dataset and reinvestigated the evolution of this toxin in this fascinating species. Similarly, coral snakes are a lineage that exhibits unique feeding ecology. Many species of coral snakes are known for their dietary specialisation, as they may chiefly feed on other snakes [38,39,40,41,42], lizards [38,43], or amphibians [38,43].

Sea kraits represent a fascinating case of colonisation of marine habitats by terrestrial snakes [44,45,46]. In this novel habitat, these descendants of landlubbers would encounter prey and predatory animals with markedly different physiologies. Consequently, such ecological transitions result in distinct evolutionary selection pressures [47]. Similarly, Australian elapids are also an extremely speciose group of snakes that epitomise the influence of drastic ecological transitions on venom evolution, especially given that the entire lineage may have descended from a semi-aquatic ancestor [48]. Subsequent to the colonisation of the Australian continent by semi-aquatic elapid snakes, they have undergone adaptive radiation, giving rise to more than 160 extant species of snakes, including a lineage (the ‘true sea snakes’) that has colonised the marine environment [49]. This adaptive radiation of snake species was also accompanied by a rapid expansion of their venom arsenal [50,51]. Therefore, we classified and analysed sea krait and Australian elapid PLA_2_s as distinct groups.

### 2.3. Distinct Phylogenetic Histories of Elapid and Viperid Venom PLA_2_s

Finally, we incorporated the unique phylogenetic histories of PLA_2_s while partitioning datasets. Nucleotide datasets were assembled by retrieving 418 and 494 group I and II sequences, respectively, and phylogenetic analyses were performed to reconstruct the evolutionary histories of snake venom PLA_2_s. Bayesian phylogenetic reconstructions of group I non-pancreatic PLA_2_s retrieved two distinct lineages (Figure 1 and Appendix A). One of the lineages was primarily composed of PLA_2_ sequences from Australian elapids, including orphan clades I-III, and was paraphyletic with sea krait (*Laticauda* spp.) and sea snake (*A. eydouxii*) PLA_2_s. The second lineage contained coral snake (*Micrurus* spp.), certain Australian elapid (orphan clade IV) and Afro-Asian and Middle Eastern elapid PLA_2_s (Figure 1 and Appendix A). The group I pancreatic PLA_2_, on the other hand, was analysed separately (Appendix A). In comparison to previously reported trees [17,26], our phylogenetic reconstructions provide robust relationships for PLA_2_s across a broader taxonomic distribution with well-supported nodes. Interestingly several deeply nested groups included closely related taxa and formed reciprocally monophyletic clades. For instance, PLA_2_ toxins from the genera *Naja*, *Bungarus*, *Micrurus* and *Laticauda* occupied distinct monophyletic clades (Figure 1, Appendix A). These clustering patterns suggest that there has been a genus-specific diversification of group I PLA_2_s under the influence of organismal ecology and evolution. It should also be noted that all the aforementioned structural–functional classes of group I PLA_2_s, as well as those that were partitioned based on ecological considerations, formed distinct monophyletic clades.

Interestingly, in contrast to elapid group I PLA_2_s, group II in Viperidae snakes did not form diverse phylogenetic groups (Figure 2 and Appendix A). The phylogeny of group II PLA_2_ was also marked by limited node support, despite numerous attempts to resolve their relationships. We speculate that this is primarily because of the ancestral recruitment of group II PLA_2_s in viperids, which was followed by the convergent evolution of diverse substitutions at the catalytic site. Convergent evolution of functional residues has previously been shown to confound phylogenetic analyses of this toxin family when transcriptomic sources of data are exclusively relied upon [52]. Extensive sequence divergence and recurrent evolution of the catalytic site perhaps make it difficult to robustly determine their phylogenetic relationships. In future, when high-quality genomic data are more available, microsyntenic analyses of gene arrangements could resolve this issue (see, e.g., [53]). However, considering our inability to precisely resolve evolutionary relationships, we retained our classification of group II PLA_2_ sets based on the amino acid substitution at the catalytic site and investigated the regimes of selection pressures acting on them. A caveat of this strategy is that it may not represent the precise phylogenetic history of these snake venom PLA_2_s. However, its advantage is that it enables us to determine whether or not convergently recruited catalytic sites influence the regime of selection pressure under which these toxins evolve (see below).

### 2.4. The Impact of Structure–Function, Ecology and Evolution on the Diversification of Snake Venom PLA_2_s

To determine the effects of structural–functional constraints on PLA_2_s and the unique ecological and evolutionary histories of snakes that produce them, we employed maximum-likelihood approaches and estimated the omega (ω) ratio for each toxin group. ω represents the ratio of non-synonymous (substitutions that change the coded protein) to synonymous substitutions (changes that do not alter the coded protein) and thereby sheds light on the nature and strength of the selection regime experienced by the genes [54,55,56,57]. An omega ratio of greater, lesser, or equal to 1 is indicative of positive selection, negative selection, and neutral evolution, respectively.

Site models implemented in PAML identified several lineages experiencing the influence of positive selection (Table 2). However, the nature and strength of selection varied across snake lineages. The strongest effect of positive selection was recorded for sea kraits and *A. eydouxii*, while the lowest was estimated for the *Naja* group [ω: 1.19; positively selected (PS): 15; Figure 3; Table 2]. As previously outlined, sea kraits exemplify a drastic shift in habitat, and the observed rapid rate of evolution (ω: 2.4; PS: 26) could be a consequence of colonising a novel niche. Similarly, the stark shift in feeding ecology recorded in the marbled sea snake could potentially explain the accelerated evolution (ω: 2.34; PS: 2) of this venom protein, either as a result of the complete relaxation of constraint or the derivation of novel forms with a primarily digestive function. These findings contradict an earlier report of decelerated evolution for *A. eydouxii* PLA_2_ [37]. The limited number of positively selected sites identified in these may further corroborate the suggestion that the accelerated rate of non-synonymous mutations in these sequences may result from a relaxation of selection, rather than the role of positive selection fixing site-specific substitutions.

Similar to the sea kraits, Australian elapids are a lineage that has successfully colonised novel ecological niches. The strong effects of positive selection observed in this group (ω: 1.5 to 1.7; PS: between 18 and 35) could be a consequence of such ecological adaptations, which have occurred rapidly within this relatively young lineage [Figure 3; Table 2; [58]]. The Bayes Empirical Bayes (BEB) approach implemented in model 8 of PAML confidently identified 86 sites in Australian elapid PLA_2_s that are experiencing a strong influence of positive selection.

We also identified lineages with signatures of negative selection (ω < 1), which contradicted previous reports of ubiquitous diversifying selection pressures on snake venom PLA_2_s (Figure 3; Table 2). The evolution of elapid β-bungarotoxins (ω: 0.96; PS: 13) was strongly constrained by purifying selection pressures. Since β-bungarotoxins result from the dimerisation of PLA_2_ and Kunitz proteins, it is crucial to conserve sites that partake in this process. A similar phenomenon was observed for κ-neurotoxins, 3FTx which form covalently bonded dimeric complexes (and also originate from snakes in the genus *Bungarus*), in our previous study on the evolution of 3FTx [14]. An accelerated rate of evolution could potentially disrupt structurally important residues and drastically affect these multimeric elapid toxins. Unsurprisingly, amino acid positions that were identified in these toxins as experiencing positive selection were outside of the dimerisation region (Figure 4A). Surprisingly, despite possessing the characteristic loop, which we suspected to be a structural constraint, the pancreatic PLA_2_s were found to evolve under positive selection and possessed a large number of positively selected sites (ω: 1.37; PS: 33), indicating that such modifications are not deleterious (Figure 3; Table 2). It is interesting to note, however, that only one of the positively selected sites identified in this toxin was found in the pancreatic loop itself (Figure 4B).

Similar to β-bungarotoxins, the evolution of viperid D49 (ω: 0.83; PS: 21) and K49 (ω: 0.92; PS: 13) PLA_2_s was also found to be severely constrained by purifying selection (Figure 5). Literature suggests that the D49 viperid PLA_2_ subtype is a plesiotypic form with haemotoxic activity [52]. When viperid snakes underwent an adaptive radiation [59], their venom PLA_2_ gene may have also rapidly diversified, resulting in novel toxin forms with apotypic substitutions at the 49th position. Interestingly, these substitutions also correlate with the emergence of atypical functions such as oedematous, neurotoxic, and myotoxic activities [52]. As our understanding of the evolutionary origins, phylogenetic relationships, and ecological relevance of PLA_2_s with distinct catalytic sites remains limited, future research is required to reconstruct the causal evolutionary pathways leading to the origins of these diverse functions of group II PLA_2_ venom toxins.

Proteins acquire new functions when they are exposed to novel interaction partners. This can be conceptualised as a change in the protein’s “ecological affordances” [51]. Such changes of a protein’s *context* within a network of interactions may result from changes in gene expression pattern, inoculation to another organism (as with “exochemical” proteins such as toxins), or changes in functional residues [1]. Rapidly diversifying surface-exposed residues is a primary way in which venom proteins acquire novel functions [14]. In keeping with this general pattern, PLA_2_s alter their surface chemistry to acquire novel molecular targets and consequently elicit distinct pharmacological effects [18]. The estimation of solvent accessible surface area (SASA) for the amino acid chains of PLA_2_s from group I and group II sets revealed that a large number of positively selected sites were surface exposed (Table 2), corroborating the results of previous reports [18]. Australian elapid PLA_2_s contribute to diverse pathologies by acting on a variety of molecular targets. Such a diverse array of activities may have arisen from the evolutionary tinkering of surface chemistry (60 surface exposed sites), while keeping the core intact to maintain structural stability (five buried sites). Sea krait and *Micrurus* PLA_2_s too exhibited a similar trend, with a larger number of positively selected sites being surface exposed than buried (exposed: 16 and 15; buried: 3 and 1 sites, respectively). Similarly, all other PLA_2_ sets were characterised by a great prevalence of positively selected sites amongst solvent accessible regions of the molecule, and very few such sites were buried within the core. Unlike all other PLA_2_ subgroups, those produced by *A. eydouxii* were only characterised by a single positively selected solvent accessible site, providing further support to the hypothesis that their accelerated evolution may result from a loss of venom function.

Furthermore, to understand whether the selection pressures driving the evolution of PLA_2_s are episodic or pervasive in nature, we employed the Mixed Effect Maximum Likelihood Model (MEME) and Fast Unconstrained Bayesian AppRoximation (FUBAR) approaches. In contrast to the site selection analyses in PAML that revealed a strong influence of purifying selection, MEME identified the highest number of episodically diversifying sites for the viperid D49 group (42 sites; Table 2). FUBAR detected 25 amino acid sites under the pervasive influence of positive selection while detecting twice as many sites experiencing the pervasive effects of negative selection. Viperidae snakes are believed to have undergone adaptive radiation nearly 30 million years ago (MYA), which was followed by a marked decrease in the speciation rate [59]. The large number of episodically diversifying sites identified here is perhaps indicative of the changes in group II PLA_2_s that occurred during the early diversification of these snakes. However, for certain other viperid PLA_2_s, very few sites were identified that evolved either under episodic or pervasive diversifying selection (K49: 12 and 13; N49: 6 and 17), which could be a consequence of the recent evolution of these forms. MEME failed to identify any such sites for the viperid S49 dataset, while FUBAR identified a single site with pervasive effects of positive selection. The fact that only model 8 of PAML was able to identify this toxin as experiencing diversifying selection could suggest that this toxin form may have begun to diversify only recently.

In congruence with the outcomes of PAML analyses, both MEME and FUBAR identified numerous episodically and pervasively diversifying sites in Australian elapid PLA_2_ genes (39 and 54 sites, respectively); a lineage of snakes that has recently undergone adaptive radiation. Several diversifying sites (episodic and pervasive) were also detected in *Micrurus* spp. (33 and 25 sites, respectively). Novel dietary adaptations may have been responsible for the diversifying effects of selection observed in these snakes. β-bungarotoxins and pancreatic PLA_2_s were also found to have episodic (23 and 21, respectively) and pervasively (14 and 32, respectively) diversifying sites, corroborating the results from PAML analyses, despite the latter analyses indicating that the genes as a whole are constrained by negative selection (see above). Surprisingly, despite the relatively recent shift in habitat for sea kraits, or the dietary specialisation in *A. eydouxii*, MEME identified a single episodically diversifying site in the former and failed to detect any episodically diversifying sites in the PLA_2_s of *A. eydouxii.* As observed in MEME, FUBAR also detected only five pervasively diversifying sites and a single site under the pervasive effects of purifying selection for the sea krait PLA_2_s. A similar trend was observed in *Aipysurus* PLA_2_s, wherein FUBAR failed to identify any sites evolving under pervasive purifying selection and identified a single pervasively diversifying site.

## 3. Conclusions

The results of the present study are in accordance with the conclusions of a recent investigation of toxin ‘recruitment’ [23] in that our analyses unequivocally demonstrate that there is no ‘one size fits all’ model of toxin evolution, even within a gene family. The impact of structural constraints and the mechanism of action on the evolution of venom toxins has also been documented in 3FTx [14], another major snake venom toxin superfamily, as well as in several other toxins in diverse taxa [47]. On the face of it, this is unsurprising, but it highlights the limitations of abstract models that do not take context—be it ecological, lineage-specific, molecular (in terms of interaction partners), ‘genomic neighbourhood’ (revealed by microsyntenic analyses), etc.—into account [1,47]. Our findings strongly indicate, in contrast to previous claims in the literature, that diversification via positive selection is not the canonical mode of evolution for all PLA_2_s and that certain lineages in fact evolve under the influence of purifying selection. The unique evolutionary histories of snake lineages, their ecological adaptations, and structural–functional constraints on the snake venom proteins, all contributing to shaping the venom arsenal [47]. By partitioning our datasets in ways that take these contextual factors into account, we have been able to use standard analytical methods to reveal hitherto undocumented complexity in the evolutionary patterns within this widespread, functionally and clinically important, and well-studied toxin family. Our work thus accentuates the importance of investigating venoms through the lens of evolutionary ecology.

## 4. Material and Methods

### 4.1. Dataset Assembly and Phylogenetic Reconstructions

Nucleotide datasets for snake venom PLA_2_ genes were assembled by performing iterative BLAST searches against National Center for Biotechnology Information’s ‘non-redundant’ (NCBI-NR) database: http://www.ncbi.nlm.nih.gov; accessed on 1 November 2019. Acquired sequences were manually curated and aligned using MUSCLE [60]. A Bayesian phylogenetic framework was employed to reconstruct the evolutionary history of snake venom PLA_2_s. The model for sequence evolution was determined using the ModelFinder program within IQTree [61]. The analysis was performed using MrBayes [62] with GTR + I+G as the preferred model for sequence evolution. Four MonteCarlo chain simulation runs were parallelised across twelve chains to execute the analysis. To terminate the analysis, a convergence diagnostic—standard deviation of split frequency (sdsf) of 0.01 was predefined, and model parameters and trees were sampled every 100th generation. Since the trees never converged for the viperid PLA_2_s despite multiple runs, the sdsf was reduced to 0.05. Post-simulation, the initial 25% of the sampled trees and their corresponding parameters were discarded as ‘burn-in’. The final topography and node supports (Bayesian posterior probability) were estimated by generating a majority-rule consensus tree using the data retained after burn-in. Subsequently, trees were visualised in Figtree [63].

### 4.2. Selection Analyses

The nature of selection pressures acting on the snake venom PLA_2_s was determined using maximum-likelihood based site-selection models implemented in CodeML binaries within the PAML package [64]. The ratio of non-synonymous (nucleotide changes that alter the coded protein) to synonymous (nucleotide changes that do not change the coded protein) substitutions, also known as ‘ω’, was estimated to identify the signatures of selection. The statistical significance of the outcomes was tested by performing a likelihood ratio test (LRT) for the nested models: M7 (null model) and M8 (alternate model). Amino acid sites evolving under the influence of positive selection were identified using the Bayes Empirical Bayes (BEB) method in M8 [65]. The episodic and pervasive effects of natural selection were determined using the Mixed Effect Model of Evolution (MEME) [66], and the Fast Unconstrained Bayesian AppRoximation (FUBAR) [67] packages hosted on the Datamonkey web server [68].

### 4.3. Structural Analysis

The effects of natural selection on snake venom PLA_2_s were visualised by generating 3D homology models for various representative sequences using the Phyre2 web server [69]. The evolutionary conservation among various amino acid sites was mapped on the crystal structures using the Consurf web server [70]. PyMOL (Schrodinger, LLC, USA 2010. The PyMOL Molecular Graphics System, Version 2.5.2) was used for the visualisation of homology models. GETAREA web server was used to estimate the solvent accessible surface area (SASA) for the amino acid chains [71]. Additionally, the Adaptive Poisson–Boltzmann Solver (APBS) plug-in was used for determining the electrostatic potential of the solvent accessible area [Connolly surface [72,73]].

## Figures and Tables

**Figure 1 toxins-14-00420-f001:**
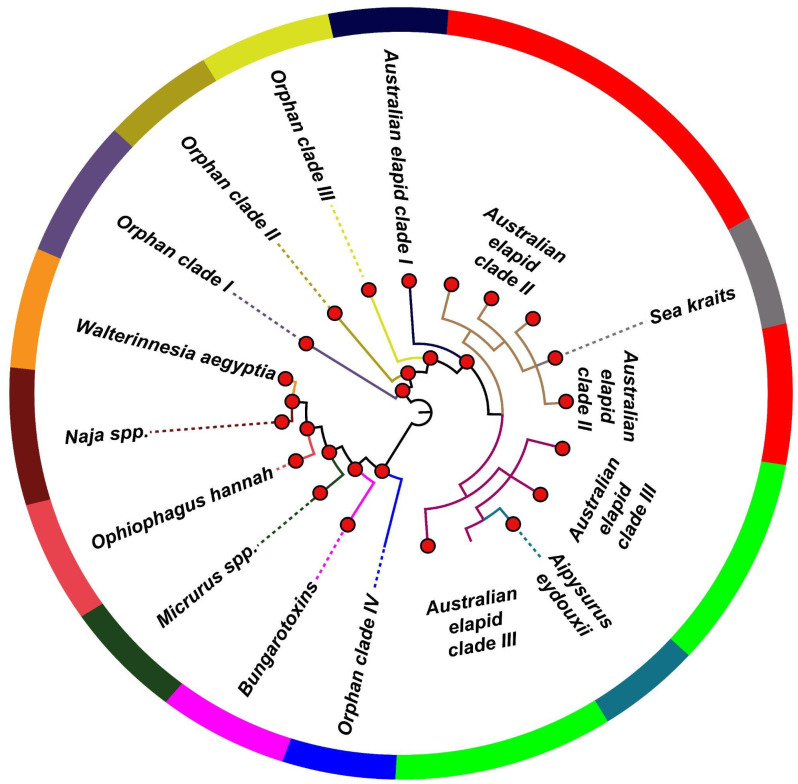
Topology of phylogenetic relationships among group I non-pancreatic PLA_2_s. A solid red circle represents node support [Bayesian posterior probability (bpp) ≥0.9].

**Figure 2 toxins-14-00420-f002:**
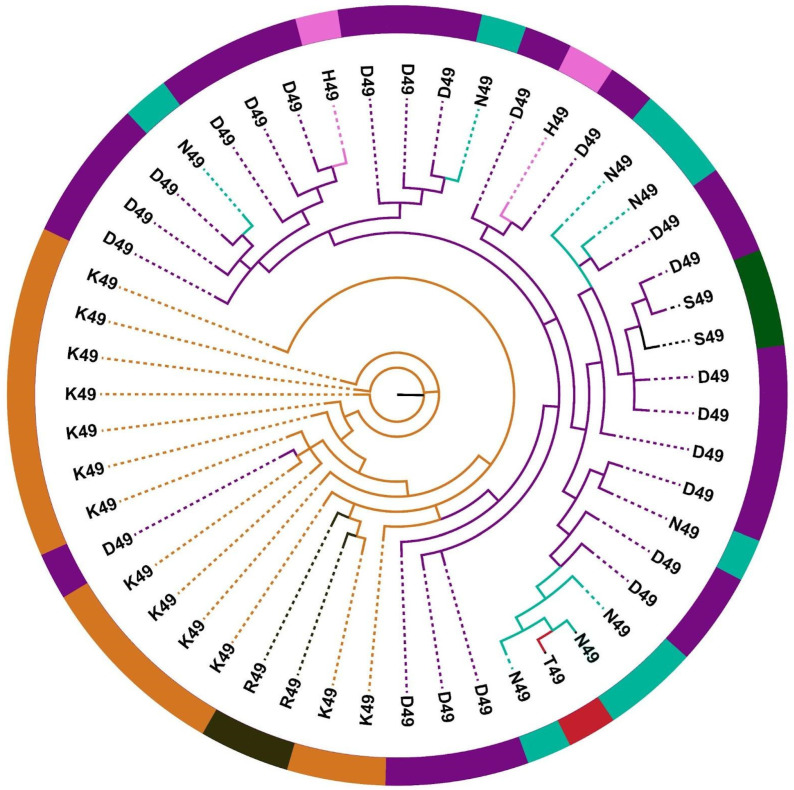
Topology of phylogenetic relationships among group II PLA_2_s.

**Figure 3 toxins-14-00420-f003:**
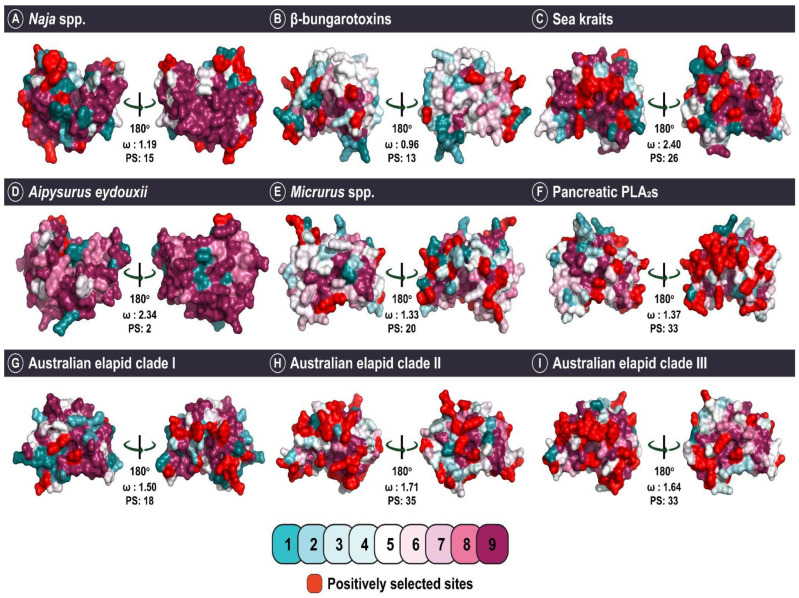
This figure portrays three-dimensional (3D) homology models depicting the influence of diverse ecologies, structural–functional constraints and evolutionary histories of snake lineages on the evolution of group I PLA_2_s. The colour-coded scale indicates the strength of evolutionary selection on these PLA_2_s, shown on a scale of 1 (highly variable) to 9 (highly conserved). Positively selected sites are shown in red.

**Figure 4 toxins-14-00420-f004:**
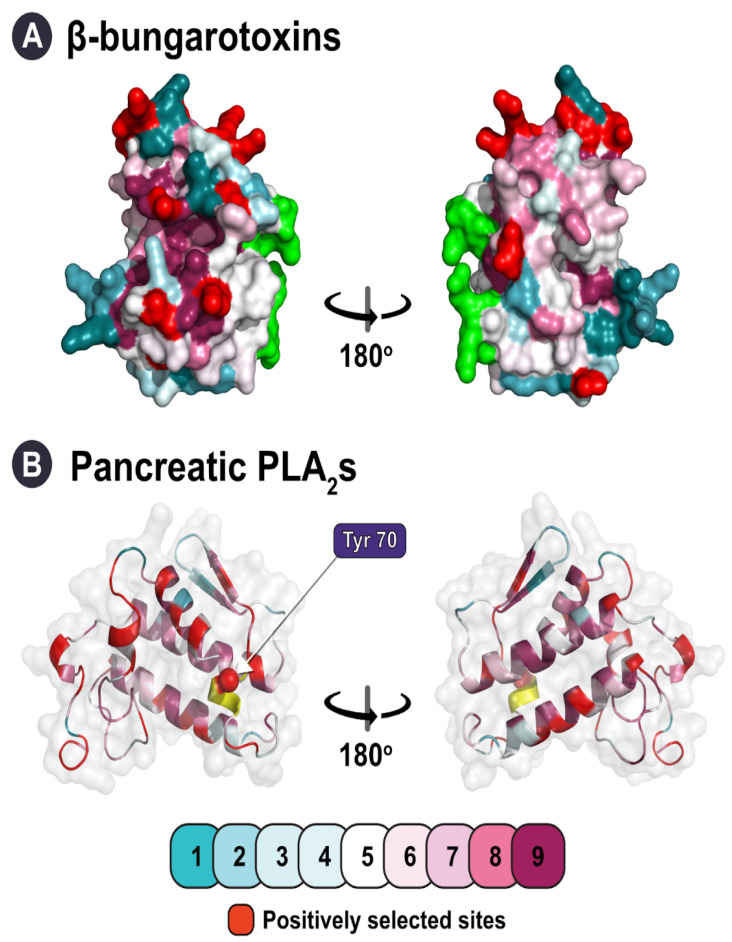
This figure depicts the impact of structural constraints on the evolution of group I PLA_2_s. Here, panel (**A**) shows a homology model of β-bungarotoxins, highlighting the regions of dimerisation (green shade), while the characteristic loop in pancreatic PLA_2_s is highlighted in panel (**B**) (yellow shade). The colour code provided indicates the strength of evolutionary selection acting on PLA_2_s on a scale of 1 (highly variable) to 9 (highly conserved). Sites identified to be under the significant influence of positive selection are shown in red.

**Figure 5 toxins-14-00420-f005:**
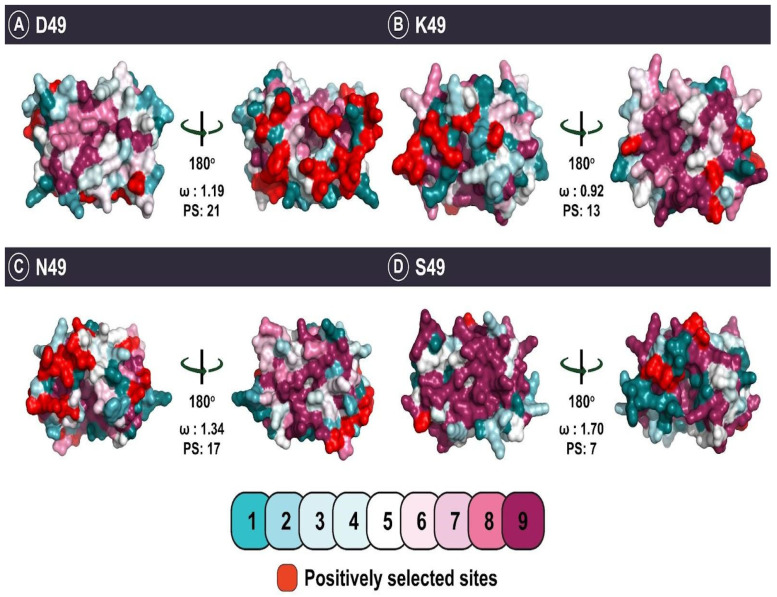
This figure depicts homology models portraying the evolution of group II PLA_2_s. The colour-coded scale indicates the strength of evolutionary selection on these PLA_2_s shown on a scale of 1 (highly variable) to 9 (highly conserved). Positively selected sites are shown in red.

**Table 1 toxins-14-00420-t001:** Schema for the classification of snake venom PLA_2_s.

	Feature
** Group I (Elapidae) **	
***Naja* spp.**	Phylogenetic clustering
** *Ophiophagus Hannah ** **	Phylogenetic clustering, ecological constraint
** *Walterinnesia aegyptia ** **	Phylogenetic clustering, ecological constraint
**β-bungarotoxins**	Phylogenetic clustering, structural constraint
**Australian elapid clades I, II and III**	Phylogenetic clustering, ecological constraint
**Sea kraits**	Phylogenetic clustering, ecological constraint
** *Aipysurus eydouxii* **	Phylogenetic clustering, dietary specialisation
***Micrurus* spp.**	Phylogenetic clustering, dietary specialisation
**Orphan clades I, II, III and IV ***	Phylogenetic clustering
**Pancreatic**	Structural constraint
** Group II (Viperidae) **	
**D49**	Catalysis
**H49 ***	Catalysis
**K49**	Catalysis
**N49**	Catalysis
**R49 ***	Catalysis
**S49**	Catalysis
**T49 ***	Catalysis

Sets excluded from selection analyses due to limited representation are marked with an asterisk (*).

**Table 2 toxins-14-00420-t002:** Molecular evolution of snake venom PLA_2_s.

Group	FUBAR ^a^	MEME ^b^	PAML ^c^	Solvent Accessibility (Number of Residues)	
				Exposed	Buried
**Elapidae**					
***Naja* spp.**	ω > 1 ^d^: 3 ω < 1 ^e^: 4	1	15	12	0
			ω **: 1.19		
**β-bungarotoxin**	ω > 1 ^d^: 14 ω < 1 ^e^: 11	23	13	9	0
			ω **: 0.96		
**Australian elapid clade I**	ω > 1 ^d^: 6 ω < 1 ^e^: 4	1	18	12	0
			ω **: 1.50		
**Australian elapid clade II**	ω > 1 ^d^: 24 ω < 1 ^e^: 10	17	35	27	2
			ω **: 1.71		
**Australian elapid clade III**	ω > 1 ^d^: 24 ω < 1 ^e^: 9	21	33	21	3
			ω **: 1.64		
**Sea kraits**	ω > 1 ^d^: 5 ω < 1 ^e^: 1	1	26	16	3
			ω **: 2.40		
** *A. eydouxii* **	ω > 1 ^d^: 1 ω < 1 ^e^: 0	0	2	1	1
			ω: 2.34		
***Micrurus* spp.**	ω > 1 ^d^: 25 ω < 1 ^e^: 6	33	24	15	1
			ω **: 1.34		
**Pancreatic PLA_2_s**	ω > 1 ^d^: 32 ω < 1 ^e^: 13	21	33	21	1
			ω **: 1.37		
**Viperidae**					
**D49**	ω > 1 ^d^: 25 ω < 1 ^e^: 50	42	21	16	0
			ω **: 0.83		
**K49**	ω > 1 ^d^: 13 ω < 1 ^e^: 10	12	13	9	1
			ω **: 0.92		
**N49**	ω > 1 ^d^: 17 ω < 1 ^e^: 8	6	17	11	1
			ω **: 1.34		
**S49**	ω > 1 ^d^: 1 ω < 1 ^e^: 0	0	7	5	0
			ω **: 1.7044		

**Legend**: **a**: Fast Unconstrained Bayesian AppRoximation to determine pervasive effects of selection; **b**: Sites experiencing episodic diversifying selection (*p* ≤ 0.01) based on the Mixed Effects Model Evolution (MEME); **c**: Positively selected sites detected by the Bayes Empirical Bayes approach implemented in M8; **d**: Number of sites under pervasive diversifying selection at the posterior probability ≥0.95 (FUBAR); **e**: Number of sites under pervasive purifying selection at the posterior probability ≥0.95 (FUBAR); ω: mean dN/dS (ω **: *p* ≤ 0.01).

## Data Availability

All the requisite data are provided as Appendix A.

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
