# Peer review of "Contextual Constraints: Dynamic Evolution of Snake Venom Phospholipase A2"

_toxins, 2022, doi:10.3390/toxins14060420_

Round 1

Reviewer 1 Report

The manuscript entitled “Contextual constraints: dynamic evolution snake venom phospholipase A2” provides novel insights regarding the evolution of the phospholipase A2 (PLA2) family, toxins found ubiquitously in snake venoms with a broad a diverse spectrum of biological functions, that make them one of the most medically important and fascinating toxin families found in the secretion of these animals. The authors employed several specialized analytical tools to obtain a comprehensive picture of the different factors involved in the evolutionary patterns seen within the PLA2s.

The manuscript is well written, allowing non-evolutionary experts to understand how the unique evolutionary histories of snake lineages, their ecological adaptations, and structural-functional constraints led to the diversity seen nowadays in this toxin family. The data displayed brings novel information that set the starting point to future analysis and research.

My comments are basically about text editing. I found the introduction lengthy, the authors can summarize this information. In addition, I got confused with the first paragraph of point 2.3 (Distinct phylogenetic histories of elapid and viperid venom PLA2s). The authors mentioned the classification of these toxins based on their structural similarities to mammalian pancreatic and synovial PLA2s, into groups I and II; however, in the paragraph mentioned above, when the authors explained the phylogenetic analysis performed (Line 188), there is a group I of non-pancreatic PLA2s that was not stated before. Can you please explain this statement?

Last but not least, I was not able to find the supplementary data (the main manuscript was downloaded twice). This complementary information needs to be revised, to obtain the final approval for publication.

Author Response

The manuscript entitled “Contextual constraints: dynamic evolution snake venom phospholipase A2” provides novel insights regarding the evolution of the phospholipase A2 (PLA2) family, toxins found ubiquitously in snake venoms with a broad a diverse spectrum of biological functions, that make them one of the most medically important and fascinating toxin families found in the secretion of these animals. The authors employed several specialized analytical tools to obtain a comprehensive picture of the different factors involved in the evolutionary patterns seen within the PLA2s

The manuscript is well written, allowing non-evolutionary experts to understand how the unique evolutionary histories of snake lineages, their ecological adaptations, and structural-functional constraints led to the diversity seen nowadays in this toxin family. The data displayed brings novel information that set the starting point to future analysis and research.

We thank the reviewer for their kind words.

My comments are basically about text editing. I found the introduction lengthy, the authors can summarize this information.

We believe that the introduction, in its current form, provides an appropriate context to the study and a rationale of the work. We, therefore, would like to retain the introduction in its current form.

In addition, I got confused with the first paragraph of point 2.3 (Distinct phylogenetic histories of elapid and viperid venom PLA2s). The authors mentioned the classification of these toxins based on their structural similarities to mammalian pancreatic and synovial PLA2s, into groups I and II; however, in the paragraph mentioned above, when the authors explained the phylogenetic analysis performed (Line 188), there is a group I of non-pancreatic PLA2s that was not stated before. Can you please explain this statement?

We thank the reviewer for pointing out this inconsistency. The group I PLA2s are further subclassified into pancreatic and non-pancreatic sequences based on the presence or absence of the pancreatic loop. This has now been clarified in the introductory text (line number: 85).

Last but not least, I was not able to find the supplementary data (the main manuscript was downloaded twice). This complementary information needs to be revised, to obtain the final approval for publication.

We thank the reviewer for highlighting the technical error. All supplemental data associated with this manuscript were submitted. We are unsure why these were not accessible. We have now re-uploaded these files.

Reviewer 2 Report

There were several concerns.

Chap 1, line 82

Not only group II, group I has also recruited as a toxin in the elapid snake lineage as a result of the co-option of a gene copy that encoded the pancreatic digestive enzyme. Then, they have lost the peptide of pancreatic loop. Biosci Biotechnol Biochem. 2013;77(1):97-102. doi: 10.1271/bbb.120595.

Chap 2.3, line 188

The two major lineages in Figure 1 may represent the differences between Australian New World cobras and other Old World cobras.

On that basis, it seems more noteworthy that neither coral snakes nor sea snakes have been displaced from the Australian cobra clades, respectively, despite such changes in diet or habitat.

Chap 2.4, line 304

Previous studies have reported that the numbers do not necessarily indicate positive selection when compared within the same group of catalytic sites such as D49 and K49. Toxicon. 2003 Dec 15;42(8):841-54. doi: 10.1016/j.toxicon.2003.11.003.

          line 219

Taken together, it is not unreasonable to assume that the speciation of snake family occurred after the recruitment of edema-inducing, hemolytic, neurotoxic D49 and myonecrotic K49, etc., rather than assuming that convergent evolution of catalytic sites occurred simultaneously after the speciation of the snake family.

          line 417

Pancreatic PLA2 is not considered to be subject to positive selection because it is a constitutive protein that has nothing to do with toxicity, and some toxic proteins, such as bungarotoxin, are not subject to positive selection.

Chap 3, line 443

There are previous studies that have partially clarified the synteny of PLA2 genes. Gene. 2010 Aug 1;461(1-2):15-25. doi: 10.1016/j.gene.2010.04.001. 

The consideration that the ecological adaptations of snakes and the structural-functional constraints on the snake venom proteins constrain the mutational tendencies did not seem to be so novel.

Author Response

Chap 1, line 82

Not only group II, group I has also recruited as a toxin in the elapid snake lineage as a result of the co-option of a gene copy that encoded the pancreatic digestive enzyme. Then, they have lost the peptide of pancreatic loop. Biosci Biotechnol Biochem. 2013;77(1):97-102. doi: 10.1271/bbb.120595.

We thank the reviewer for highlighting the work by Chijiwa et al. 2013. We believe that the reviewer wants to highlight the presence of pancreatic group I PLA2s (that are generally found in elapids) in Viperidae, and not elapids as they say in their comment. While this finding is intriguing, the role of convergent evolution cannot be ruled out as similar orthologs have not been reported from the venoms of other viperid snakes. Moreover, since Chijiwa et al. 2013 have recovered PLA2s from liver tissues via PCR and not from venom glands (i.e., they may have simply recovered physiological counterparts and not toxic PLA2s), this does not provide evidence against the independent recruitment of group I and II PLA2s in elapid and viperid snakes, respectively.

Chap 2.3, line 188

The two major lineages in Figure 1 may represent the differences between Australian New World cobras and other Old World cobras. On that basis, it seems more noteworthy that neither coral snakes nor sea snakes have been displaced from the Australian cobra clades, respectively, despite such changes in diet or habitat.

We did not understand this comment from the reviewer. There are no ‘new world cobras’ or cobras in Australia. In any case, gene trees do not always reflect on species relationships. The fact that the Australian PLA2 clade in itself is broken down into seven subclades highlights the influence of contrasting evolutionary/phylogenetic histories, in line with our hypothesis.

Chap 2.4, line 304

Previous studies have reported that the numbers do not necessarily indicate positive selection when compared within the same group of catalytic sites such as D49 and K49. Toxicon. 2003 Dec 15;42(8):841-54. doi: 10.1016/j.toxicon.2003.11.003.

The KA/KS estimates by Ohno et al. 2003 were based on pairwise comparisons of a limited number of sequences that were available at the time. It is now very clear that pairwise comparisons are not sensitive and unreliable (Yang and Bielawski 2000).

line 219

Taken together, it is not unreasonable to assume that the speciation of snake family occurred after the recruitment of edema-inducing, hemolytic, neurotoxic D49 and myonecrotic K49, etc., rather than assuming that convergent evolution of catalytic sites occurred simultaneously after the speciation of the snake family.

We would like to bring it to the reviewer’s attention that none of the forms have been directly recruited into the venom/venom gland of snakes. These diverse forms have evolved subsequent to the recruitment and weaponisation of the PLA2 gene into the venom glands of snakes. Both the hypotheses, i.e., the MRCA of vipers having these diverse PLA2 forms in its venom or the convergent evolution of these forms subsequent to speciation, are plausible. However, due to the lack of evidence supporting the first hypothesis, we have only discussed the convergent evolution of these forms. Furthermore, we have referred to the Malhotra et al. 2015 study only to highlight that the use of transcriptomic sequences would not suffice to determine the complex processes that have underpinned the evolution of these diverse snake venom PLA2 activities.

line 417

Pancreatic PLA2 is not considered to be subject to positive selection because it is a constitutive protein that has nothing to do with toxicity, and some toxic proteins, such as bungarotoxin, are not subject to positive selection.

We would like to bring to the reviewer’s attention that we are referring to the snake venom pancreatic PLA2s [the ones expressed abundantly in elapid snake venoms (Huang et al. 1997, Francis et al. 1997, Fohlman et al. 1977) and not the PLA2s expressed in their pancreatic tissues]. As far as the lack of positive selection on bungarotoxin is concerned, this has already been explained in the text.

Chap 3, line 443

There are previous studies that have partially clarified the synteny of PLA2 genes. Gene. 2010 Aug 1;461(1-2):15-25. doi: 10.1016/j.gene.2010.04.001.

We would like to highlight that the work done by Ikeda et al. 2010 only explains the synteny of PLA2 genes within a small genomic region of Protobothrops flavoviridis, and has not compared these regions across diverse snake lineages and other non-serpent taxa. In-depth comparisons analysing diverse vertebrate lineages could provide novel insights into the evolution of toxin gene superfamilies and aid in understanding their diversification. We have recently investigated the synteny of phospholipase A2s across 93 vertebrate lineages to specifically address this shortcoming (Koludarov et al. 2020). However, the current phylogenetic models fail to incorporate gene synteny while determining evolutionary relationships. Addressing this caveat and improving the genomic resources available for these analysis could help in robustly tracing the evolution of different gene families

The consideration that the ecological adaptations of snakes and the structural-functional constraints on the snake venom proteins constrain the mutational tendencies did not seem to be so novel.

We respectfully disagree. As the reviewer doesn’t support this statement with citations, we are unsure which publications they are referring to. Both ecological and evolutionary constraints are always invoked in the literature without evidence. This work, in addition to serving as a robust proof for the extant postulates, presents a strong case for the necessity of investigating venom evolution in the light of the ecological contexts that underpin them. By classifying the PLA2s into various groups based on structural-functional constraints and the unique ecological and evolutionary histories of snakes, our findings offer a better resolution of the selection regimes shaping the evolution of various group I and II PLA2s.